# Controllable DNA hybridization by host–guest complexation-mediated ligand invasion

Lin Xiao[1,2], Liang-Liang Wang[1,2], Chao-Qun Wu[1], Han Li[1], Qiu-Long Zhang[1], Yang Wang[1] & Liang Xu [1] ✉

Dynamic regulation of nucleic acid hybridization is fundamental for switchable nanostructures and controllable functionalities of nucleic acids in both material developments and biological regulations. In this work, we report a ligand-invasion pathway to regulate DNA hybridization based on host–guest interactions. We propose a concept of recognition handle as the ligand binding site to disrupt Watson–Crick base pairs and induce the direct dissociation of DNA duplex structures. Taking cucurbit[7]uril as the invading ligand and its guest molecules that are integrated into the nucleobase as recognition handles, we successfully achieve orthogonal and reversible manipulation of DNA duplex dissociation and recovery. Moreover, we further apply this approach of ligand-controlled nucleic acid hybridization for functional regulations of both the RNA-cleaving DNAzyme in test tubes and the antisense oligonucleotide in living cells. This ligand-invasion strategy establishes a general pathway toward dynamic control of nucleic acid structures and functionalities by supramolecular interactions.

Nucleic acid duplex structures are formed by highly stable Watson–Crick base pairs. Dynamic regulation of nucleic acid hybridization is not only important for manipulation of nucleic acid structures and functions in biological systems but also critical for controllable nucleic acid-based nanomaterials[1–3]. In contrast to the assembly process that is typically simultaneous, direct dissociation of a thermostable DNA/RNA double helix without heating is challenging. In living systems, some DNA binding proteins are required to unwind the duplex structure[4–6]. This process requires cooperative effects from diverse proteins, but reversible control of these protein activities seems unlikely to operate in a simple and convenient manner. Competing nucleic acid strands can efficiently initiate the duplex separation via a toehold, but are incapable of direct invasion[7]. Some nucleic acid duplexes may be regulated by ligand treatments through formation of specific secondary structures (e.g. ligand–aptamer interaction, G-quadruplex and i-motif formation)[8–11], but these conversions can only work with constrained sequences. Alternatively, high

concentrations of denaturing reagents and nanomaterials[12,13], as well as intensive pH changes, can also be utilized to directly destruct DNA hybridization[14], but these treatments are indiscriminately subject to all nucleic acid structures and difficult to manipulate in a reversible approach.

Compared with direct dissociation of an unmodified nucleic acid duplex, introduction of some artificial functional groups into a modified nucleic acid strand to regulate its hybridization is a more achievable pathway. This concept was pioneered about two decays ago, when an azobenzene group was introduced into the DNA strand as a disruptor inside the duplex structure[15,16]. On the basis of light-induced trans-cis transformation of azobenzene to cause steric clash against base stacking within the double helix[17,18], DNA duplex structures can reversibly switch between dissociation and formation. With utilization of azobenzene and its derivatives, the photo-controlled regulations of DNA duplexes have been widely applied in a variety of nucleic acid-based research fields

[1]MOE Key Laboratory of Bioinorganic and Synthetic Chemistry, School of Chemistry, Sun Yat-Sen University, Guangzhou 510275, China. [2]These authors contributed equally: Lin Xiao, Liang-Liang Wang. ✉e-mail: xuliang33@mail.sysu.edu.cn

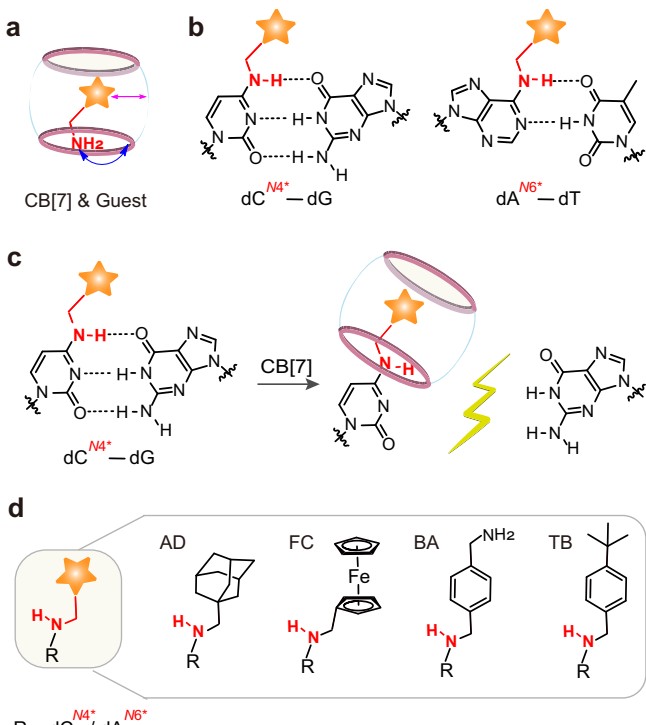

**Fig. 1 | Design of host–guest recognition-mediated invasion of Watson–Crick base pairs. a** Recognition mode of CB[7] and the guest molecule. For a typical guest molecule with an amine group, the hydrophobic component (the star symbol) enters inside the cavity, living the amine group interacting with the portal oxygen atoms through hydrogen bonding and dipolar interactions. **b** Presumably base pairing with the guest-modified nucleobases. The guest molecule is integrated into the $N^4$-position of cytosine or the $N^6$-position of adenine. **c** CB[7]-induced invasion of base pairs. Recognition of the guest group by CB[7] causes breaking of base pairs through steric clash. **d** Selected guest molecules for integration into the $N^4$-position of cytosine or the $N^6$-position of adenine as recognition handles.

involving switchable DNA structures and nanomaterials for bio-sensing and diagnostics, drug delivery, and regulation of gene expressions[19–21]. The effectiveness of the azobenzene group inspired several other approaches of photo-controlled regulations of DNA duplexes through introduction of different light-responsive groups[22–26].

Despite the great development of these photo-controlled strategies, it still lacks other general pathways to reversibly disrupt a random nucleic acid duplex structure and regulate its dissociation. The biggest challenge is, in fact, the disruption of tightly Watson–Crick formed base pairs. This process is thermodynamically disfavored and typically requires additional forces, e.g. the photo-energy in the previously reported light-induced isomerization[17,18], to dissociate the duplex structure. In nature, some DNA binding proteins can recognize modified nucleobases and thereafter employ invading residues to compete with Watson–Crick base pairs as the driving force for DNA duplex disruption[27,28]. We envisioned that whether the concept of protein invasion could be utilized for a more controllable small-ligand invasion. Could we introduce a recognition handle into the DNA duplex to mediate the competition between ligand invasion and base pairing? In this design, we expected a ligand could bind with the handle, but part of the handle was involved into Watson–Crick base pairs. As a result, once the handle was fully grabbed by the ligand, it would unavoidably block the formation of base pairs, and in turn disrupt the DNA duplex structure. Therefore, the key to this question would be what kind of recognition handle could be utilized to mediate the ligand invasion, and whether the binding with the handle could outperform the base pairing.

In this work, we turned our attention toward the host–guest chemistry, the cucurbit[7]uril-based (CB[7]) recognition due to its high binding affinity and specificity with different types of guest molecules in the aqueous solution[29,30]. Although previous studies on cucurbiturils had introduced the host–guest interactions into nucleic acid manipulation[31–40], the role of cucurbiturils did not involve direct regulation on the native structures of nucleic acids but only imposed indirect or external impacts to alter their structural conversions or functional performance. For instance, some ligands that are capable of interacting with nucleic acids can be designed to be complexed with cucurbiturils to control their effects on nucleic acid structures, such as the B-Z conversion of duplex DNA[37], formation of G-quadruplex[36] and induction of DNA condensation[31,32], but these impacts are originated from specific reagents or assemblies, whereas not directly from cucurbiturils.

Herein, we select CB[7] as the invading ligand and integrate some well-studied guest molecules into the nucleobases as recognition handles to construct a ligand-invasion pathway for controllable DNA hybridization based on host–guest interactions. Our investigations demonstrate the effectiveness of our design, and successfully achieve reversible and orthogonal regulation of DNA hybridization by chemical ligands. Moreover, we further apply this strategy in functional DNA structures and achieved conditional control of DNA functionalities in both test tubes and living cells. This study describes a universal approach that can reversibly and specifically control nucleic acid hybridization without any sequence constraints through treatments of chemical ligands. Our development of ligand-controlled DNA hybridization provides a general pathway for structural regulation of nucleic acids, and can further enrich the toolbox for manipulation of nucleic acid structures and functionalities in both material and biomedical applications.

## Results
### Design and synthesis
To achieve effective invasion of DNA base pairs, the guest molecules need to be integrated into the nucleobases with overlapped interactions, and in the meantime, must not prevent the formation of stable DNA duplex structures. Analysis of the interaction mode between CB[7] and its guest molecules revealed an important structure feature. As depicted in Fig. 1a, some potent guests generally require the recognition of an amine group or its cation derivatives by the oxygen atoms located in the portal of CB[7] via hydrogen bonding and dipolar interactions[41]. Given the amine group is also involved with hydrogen bonding in the base pairs, we envisioned that this mutual group might be utilized to function as an overlapped site during the ligand invasion. Hence, we introduced the guest molecules into the 4'-amino position of cytosine and the 6'-amino position of adenine (Fig. 1b), which could still possess the base pairing behavior. Besides, the integrated guest molecules are located into the major groove of DNA duplex, which typically has less impacts on the structural stability of DNA hybridization. With this design of recognition handle, we anticipated the binding on the integrated guest by CB[7] would cause a dramatic disruption of DNA base pairs due to the competitive recognition and the inevitable steric clash (Fig. 1c).

To systematically characterize the effect of recognition handle and dissect the ligand-invasion behavior, we selected four guest molecules (Fig. 1d), 1-adamantanemethylamine (AD), ferrocenyl methylamine (FC), 1,4-benzenedimethanamine (BA) and 4-tert-butylbenzylamine (TB), with a wide range of binding affinities on CB[7] ($K_a$ values of previously reported cation derivatives of these guest molecules were -$10^{14}$ M$^{-1}$ for AD[42], -$10^{12}$ M$^{-1}$ for FC[43], -$10^9$ M$^{-1}$ for BA[44] and -$10^6$ M$^{-1}$ for TB[45]). Since our following investigations were mainly performed in the buffer with a nearly neutral pH (=7.2) and high concentrations of sodium ions (>100 mM), which was different from the previous conditions, we then measured the binding constants

**Fig. 2 | Synthesis of guest-containing nucleosides. a** TPS, DIPEA, DCM, 0 °C; (**b**) R-NH₂, DIPEA, DMF, 90 °C; (**c**) TBAF, THF, room temperature; (**d**) DMTr-Cl, pyridine, room temperature; (**e**) (OCH₂CH₂CN)(iPr₂N)PCl, DIPEA, DCM, 0 °C; (**f**) BOP, THF, DIPEA, room temperature; (**g**) R-NH₂, Cs₂CO₃, DME, room temperature. TBDMS tert-butyldimethylsilyl, TPS 2,4,6-triisopropylbenzenesulfonyl chloride, DIPEA N,N-diisopropylethylamine, TBAF tetrabutylammonium fluoride, DMTr 4,4′-dimethoxytriphenylmethyl, DMAP 4-dimethylaminopyridine, BOP 1H-benzotriazol −1-yloxytris(dimethylamino)phosphonium hexafluorophosphate, DME 1,2-dimethoxyethan. Details are provided in Supplementary Information.

between CB[7] and these guest molecules under our experimental solution. As shown by the isothermal titration calorimetry (ITC) analysis (Supplementary Fig. 1), the binding constants between CB[7] and these four guest molecules were $1.64 \times 10^9 \text{M}^{-1}$ for AD, $4.70 \times 10^8 \text{M}^{-1}$ for FC, $8.88 \times 10^5 \text{M}^{-1}$ for BA and $3.98 \times 10^4 \text{M}^{-1}$ for TB, which were significantly reduced mostly due to the effects of pH and the interactions between sodium cations and the carbonyl portals of CB[7]. Even though, these four guest molecules still presented a similar trend of binding affinities with an adequate range on CB[7].

Synthesis of these modified nucleosides with integration of guest molecules (dC$^{N4*}$ and dA$^{N6*}$) were carried out through two different approaches from uridine and inosine, respectively (Fig. 2). For synthesis of guest-modified cytidine, the TBDMS-protected uridine was first conjugated with the triisopropylbenzenesulfonyl activation group at the O6 position, which was then followed by reactions with nucleophilic amine compounds to obtain guest-containing derivatives (Fig. 2, Path A). Synthesis of guest-modified adenosine was performed through a more facile approach in which the deoxyribose was not necessary to be protected during conjugation of these amine compounds (Fig. 2, Path B) as reported previously[46]. These modified cytidine and adenosine nucleosides were incorporated into oligodeoxynucleotides (ODNs) by the conventional phosphoramidite method via the solid phase synthesis. All modified ODNs were purified through HPLC and characterized by mass spectra.

**Effects of ligand invasion**
To investigate the role of these recognition handles during ligand invasion, we first generated 15-nt ODNs with a single site of modified cytosine (Supplementary Table 1). ITC analysis indicated CB[7] still specifically bound with the AD ($K_a = 1.31 \times 10^7 \text{M}^{-1}$) and FC ($K_a = 2.95 \times 10^6 \text{M}^{-1}$) groups in the context of DNA strand but exhibited feeble affinities on the BA and TB groups (Supplementary Fig. 2), which followed a similar trend as the free guest molecules. In addition to the modified cytosine, we also generated the same sequence of ODN with one site of modified adenine and determined the binding constants between CB[7] and this guest-integrated ODN (Supplementary Fig. 3). Consistently, CB[7] prefers to bind with the AD ($K_a = 1.34 \times 10^6 \text{M}^{-1}$) and FC ($K_a = 7.44 \times 10^5 \text{M}^{-1}$) groups of the modified adenine rather than the other two. Notably, the binding affinities with these integrated guest

molecules were relatively weaker than their free states, potentially attributed to steric hindrance by the nucleobase.

If CB[7] can invade into DNA duplex structures via these integrated guest molecules, a direct consequence would be disruption of the duplex thermostability. We determined ΔT$_m$ values of the 15-bp duplex with one guest-modified cytosine in the presence of different concentrations of CB[7] (Table 1 and Supplementary Fig. 4). For the unmodified DNA, treatment of CB[7] did not alter the duplex stability (Supplementary Fig. 5). In the presence of an AD-modified cytosine, addition of CB[7] greatly reduced the duplex thermostability by -9.6 °C even with a 1 μM concentration, indicating a great effectiveness of this recognition handle during the disruption of duplex structure. For the FC-containing duplex, treatment of CB[7] also sharply weakened the stability of the duplex structure, but in this case, higher concentrations of CB[7] were needed to boost its destabilizing effects (−8.5 °C with 300 μM CB[7]). In contrast, with introduction of the BA and TB-modified sites, the CB[7]-induced impacts were much weaker than the above two. These duplex disruption activities were fully consistent with the intrinsic binding affinities between CB[7] and free guest molecules, revealing that a potent recognition was needed to compete with the base pair. These destabilization effects together with binding analysis suggested CB[7] indeed disrupted the duplex structure through these guest-mediated recognition handles, provided that the host–guest interactions were potent enough.

In addition to the modified cytosine, we also investigated the effects of the modified adenine with integration of these guest molecules. As described in Supplementary Table 2 and Supplementary Fig. 6, treatment of CB[7] could also weaken thermostability of the duplex DNA in the presence of guest-modified adenine, and the more potent the host–guest binding was, the more disruptive towards the duplex structure, which followed the similar trend as the cytosine modifications. These data further demonstrated the effectiveness of our design of recognition handle.

Having showed invasion of CB[7] via one recognition handle caused significant disruption against the thermostability of duplex DNA, we then examined whether introduction of more invading sites would multiply the effects. We designed a 19-nt ODN with two modified cytosine sites (Table 1). Indeed, addition of CB[7] into the modified duplex DNA with two recognition handles induced more potent

**Table 1 | Melting temperatures (T_m) of guest-modified DNA duplexes treated by different concentrations of CB[7]**

| ODN[a] | 5′- ACACTGT C* ACACTGC -3′ | | | | | | | | 5′- CGATGA C* TGAGCA C* TTCGT -3′ | | | | | | | |
|---|---|---|---|---|---|---|---|---|---|---|---|---|---|---|---|---|
| | AD | | FC | | BA | | TB | | AD | | FC | | BA | | TB | |
| CB[7] (µM) | T_m[b] | ΔT_m | T_m | ΔT_m | T_m | ΔT_m | T_m | ΔT_m | T_m | ΔT_m | T_m | ΔT_m | T_m | ΔT_m | T_m | ΔT_m |
| 0 | 50.4 | – | 52.9 | – | 56.4 | – | 50.4 | – | 55.2 | – | 55.5 | – | 58.9 | – | 52.7 | – |
| 1 | 40.8 | -9.6 | 52.9 | 0 | 56.4 | 0 | 50.4 | 0 | 34.0 | -21.2 | 55.0 | -0.5 | 58.7 | -0.2 | 52.7 | 0 |
| 3 | 40.8 | -9.6 | 52.7 | -0.2 | 56.3 | -0.1 | 50.4 | -0.1 | 31.0 | -24.2 | 53.6 | -1.4 | 58.7 | -0.2 | 52.7 | 0 |
| 10 | 40.8 | -9.6 | 51.4 | -1.5 | 56.4 | 0 | 50.4 | 0 | 30.9 | -24.3 | 50.4 | -5.1 | 58.6 | -0.3 | 52.7 | 0 |
| 30 | 40.8 | -9.6 | 49.9 | -3.0 | 56.3 | -0.1 | 50.3 | -0.1 | 30.5 | -24.7 | 46.6 | -8.9 | 58.4 | -0.5 | 52.3 | -0.4 |
| 100 | 40.8 | -9.6 | 48.1 | -4.8 | 56.2 | -0.2 | 50.3 | -0.2 | 30.2 | -25.0 | 41.3 | -14.2 | 57.3 | -1.6 | 52.4 | -0.3 |
| 300 | 40.8 | -9.6 | 44.4 | -8.5 | 54.8 | -1.6 | 50.1 | -0.3 | 30.5 | -24.7 | 37.0 | -18.5 | 55.0 | -3.9 | 51.3 | -1.4 |

aDNA duplex (0.2 µM) was formed with the unmodified complementary strand in the presence of 20 mM sodium cacodylate (pH = 7.2), 100 mM NaCl and 0.1 mM EDTA.
bThe unit is °C. All Tm values were obtained from two independent replicates.

destabilization than the single one (Table 1 and Supplementary Fig. 7). With two AD-modified sites, $\Delta T_m$ values reached -25 °C even with 3 µM CB[7], which was much more dramatic than the single site. Similar behaviors were also observed from two FC-modified sites, in which $T_m$ values of the modified duplex DNA were gradually reduced from 55 °C to 37 °C with treatments from 0 to 300 µM CB[7]. Interestingly, even for the weaker recognition handles BA and TB, introduction of two modified sites also enhanced the destabilization effects. These thermostability data suggested that DNA duplex structures could be more severely disrupted by ligand invasion with introduction of multiple recognition handles.

How did the ligand invasion occur during the treatment of CB[7]? Herein, we performed a UV-based kinetic analysis to directly monitor the disruption of base pairs during CB[7] invasion. The AD-modified cytosine was selected as a representative. Compared with unmodified DNA duplexes, treatment of CB[7] into the AD-modified duplexes (15-bp with one modified site and 19-bp -with two modified sites) clearly induced hyperchromicity at 260 nm (Fig. 3), whereas either CB[7] or adamantane had no UV absorbance near this wavelength. With more binding sites, invasion through two recognition handles caused more absorbance changes than the single one (Supplementary Fig. 8). These hyperchromicities were most likely attributed to the break of Watson–Crick base pairs by CB[7] invasion, and in the meanwhile, the large size of CB[7] may also cause steric clash with the neighboring nucleosides and disrupt their base pairs. Notably, the efficiency of CB[7] invasion was concentration-dependent, and the apparent kinetic rates ($k_{app}$) were calculated to be 79 min$^{-1}$ • M$^{-1}$ for 15-bp and 230 min$^{-1}$ • M$^{-1}$ for 19-bp. Clearly, invasion of CB[7] into the 19-bp duplex was more efficient than the 15-bp duplex, which was likely caused by the cooperative effect between these two recognition handles. Moreover, the local disruption of the duplex DNA caused by ligand invasion was also observed from the weakened signals of circular dichroism spectra (Supplementary Fig. 9), further revealing destruction of DNA duplex structures. Taken together, these results suggested the invasion of CB[7] directly induced disruption of DNA base pairs through the host–guest interaction.

**Orthogonal and reversible control of DNA hybridization**
Ligand invasion into Watson–Crick base pairs established the molecular basis for structural manipulation of nucleic acid hybridizations. Analysis of the concentration-dependent $\Delta T_m$ values of the 19-bp duplex revealed a dramatically different response on the concentration effects of CB[7] between the AD and FC recognition handles, in which a low concentration of CB[7] were enough to induce a maximum disruption with the AD modification, whereas a high concentration of host molecule was needed to generate intense disruption towards the FC-modified duplex structure (Supplementary Fig. 10). This distinction was caused by discrepant binding affinities of CB[7] but established a foundation for selective and multiplex manipulation of DNA structures orthogonally via different recognition handles. We then designed two orthogonal DNA duplex structures (Duplex 1 and 2), in which one strand was incorporated with recognition handles and the other strand could form a hairpin structure once released from the duplex (Fig. 4a). Different fluorophores (FAM and TMR) were labeled for these two hairpins to separately monitor structural switches of each duplex. Both duplexes co-existed in the same system. During the titration of CB[7], Duplex-1 was firstly converted into the hairpin form as evidenced from the decreased fluorescence of FAM (Fig. 4b and Supplementary Fig. 11). Even when Duplex-1 was totally dissociated by 200 nM CB[7], Duplex-2 was still stably maintained. With further increase of the CB[7] concentration, Duplex-2 was then destructed to generate the hairpin structure. With titration of CB[7], we achieved selective dissociation of one duplex at a designated concentration, leaving the other one unaffected in the same system. This

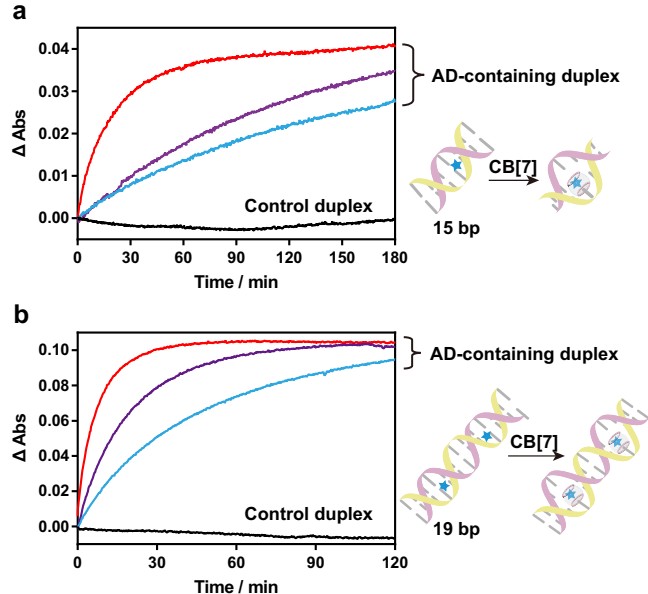

**Fig. 3 | Kinetic analysis of CB[7] invasion into the AD-containing dsDNA monitored by UV absorbance. a** CB[7] invasion into the 15-bp dsDNA with one AD-modified cytosine site. **b** CB[7] invasion into the 19-bp dsDNA with two AD-modified cytosine sites. The concentration of dsDNA was 10 μM. Three different concentrations of CB[7], 100 μM (cyan curves), 200 μM (violet curves) and 500 μM (red curves), were added into the AD-containing duplex DNA. The control duplex indicated the unmodified dsDNA with treatment of 500 μM CB[7]. Source data are provided as a Source Data file.

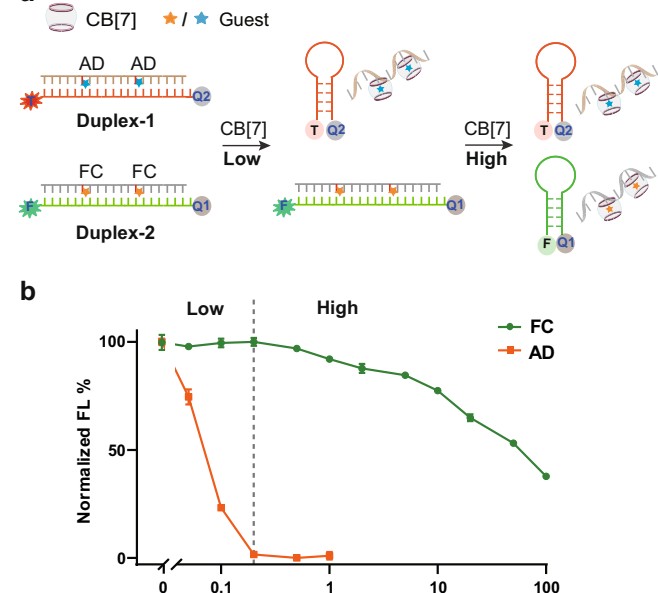

**Fig. 4 | Orthogonal control of DNA duplex dissociation by different concentrations of CB[7]. a** Schematic figure for regulation of two DNA duplexes by different concentrations of CB[7] in the same system (Duplex−1 with the AD modifications and Duplex-2 with the FC modifications). The orange star refers to the integration of FC group; the blue star refers to the integration of AD group. Different fluorophores (FAM and TMR) were labeled for these two hairpins to separately monitor structural switches of each duplex. **b** Normalized fluorescence intensities of the above system with treatment of different concentrations of CB[7]. The concentration of each duplex DNA was 10 nM. The concentrations of CB[7] were varied from 50 nM to 100 μM. Data are presented as mean values with standard deviations derived from three independent replicates. Source data are provided as a Source Data file.

concentration-dependent working style seemed to resemble the light-controlled azobenzene derivatives with specific response on different wavelengths[19,47]. Given so many different guest molecules with a wide range of binding affinities were available for CB[7] recognition, more orthogonal and multiple response of DNA duplexes would be potentially controlled in a concentration-dependent manner.

With accumulated destabilization effects through introduction of multiple recognition handles, in fact, we could directly achieve any DNA duplex dissociation induced by CB[7] at a desired temperature with a rationally designed DNA length and sequence. Herein, a fluorophore-labeled 15-nt ODN with either three AD or FC-modified cytosine sites was paired with the quencher-labeled complementary strand. Indeed, fluorescence measurements indicated that the quenched fluorescence signals were fully recovered upon the titration of CB[7] at the room temperature (Supplementary Fig. 12), demonstrating the formation of totally separated DNA strands.

Host–guest recognition is a reversible process, in which the bound molecule could be dissociated via the competitive binding. Since the invasion of CB[7] induced the duplex dissociation, removal of CB[7] by a competing guest molecule would recover the duplex formation (Fig. 5a). Taking this 15-bp duplex DNA with FC-modifications as an example (Supplementary Table 1), we performed kinetic analysis of duplex dissociation by treatment of CB[7] and its recovery by addition of the competing guest molecule FC. As depicted in Fig. 5b, the fluorescence was quenched when the duplex was formed with the complementary ODN (cODN), but was then efficiently and fully recovered to the initial single-stranded state upon the addition of CB[7] (calculated $k_{app} = 2.0 \times 10^3 \, M^{-1} \cdot s^{-1}$). The signal was then quenched again after the treatment of FC due to the recovery of the duplex structure. Moreover, cyclic additions of CB[7] and FC could control the dissociation and recovery of DNA duplex structure without significant loss of reversibility within five times (Fig. 5c and Supplementary Fig. 13). Theoretically, with introduction of proper numbers and positions of recognition handles, any random sequence of DNA

hybridization may be reversibly controlled at a desired temperature. To exclude the impacts of CB[7] or guest molecules on the fluorophore signals, we confirmed that treatments of these compounds could hardly influence the fluorescence in our experimental conditions (Supplementary Fig. 14). In addition to the FC-containing strand, similar dissociation and recovery of the duplex structure was also observed with AD as the recognition handle (Supplementary Fig. 15). Notably, compared with FC, the kinetic efficiency of AD was much slower for both invasion and recovery processes, possibly due to the slow kinetic binding/dissociation behavior between the hydrophobic adamantane and CB[7][41,48].

### Ligand-regulated functional DNA systems

Controllable reversibility of DNA hybridization provided the general basis for manipulation of functional DNA systems. Here we first took a widely investigated magnesium-based RNA-cleaving DNAzyme[49] as an example to test how this ligand invasion could control its functionality. We placed one FC-modified cytidine into each binding arm of the DNAzyme (Fig. 6a). Without addition of CB[7], the enzymatic efficiency was highly energetic; upon the treatment of CB[7], the cleavage of the substrate strand was greatly diminished due to the dissociation between the DNAzyme and the substrate strand (Fig. 6b and Supplementary Fig. 16). After further treatment with FC as the competing guest to remove CB[7], the DNAzyme activity was then reactivated. This reversible process explicitly demonstrated the controllability of DNAzyme system by ligand regulation. Moreover, real-time and reversible enzymatic regulations were also achieved during the process of cleavage (Fig. 6c and Supplementary Fig. 17). With FC-modified cytosine as the invading sites, addition of CB[7] into the ongoing

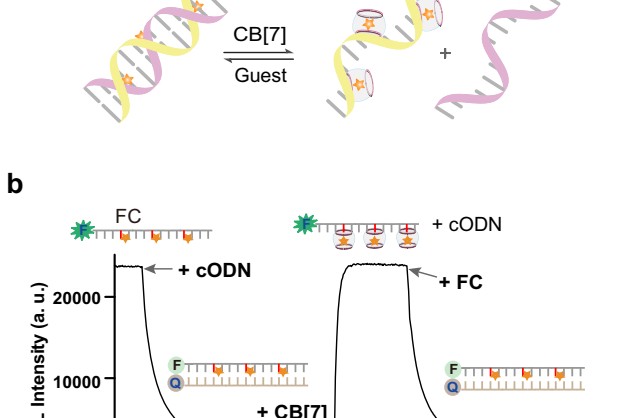

**a**

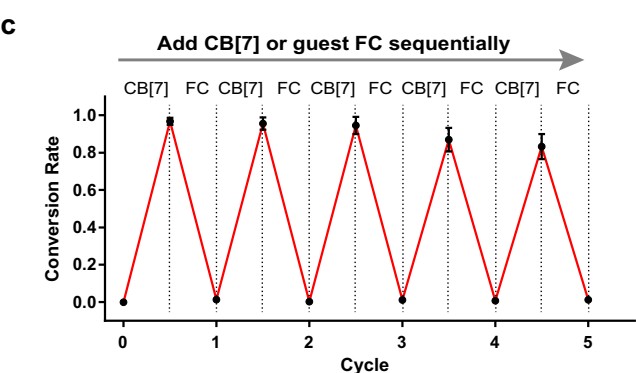

**b**

**c**

**Fig. 5 | Reversible control of DNA duplex hybridization by CB[7]. a** DNA duplex dissociation induced by treatment of CB[7] and its recovery induced by addition of competing guest molecules. **b** Kinetically monitoring of DNA duplex dissociation and formation with the integrated FC group as the recognition handle. The FC-containing strand (50 nM) was labelled by FAM and the complementary strand (cODN, 62.5 nM) was labelled by BHQ–1. 10 μM CB[7] and 30 μM FC were added during the kinetic process. **c** Conversion rates of DNA duplex dissociation upon cyclic treatments of CB[7] and the competing guest molecule FC sequentially. The concentrations of CB[7] were 5, 10, 15, 20 and 25 μM, sequentially. The concentrations of FC were 7, 12, 17, 22 and 27 μM, sequentially. Data are presented as mean values with standard deviations derived from three independent replicates. Source data are provided as a Source Data file.

system caused a timely halt in DNAzyme cleavage. Subsequent addition of FC into the suspended DNAzyme system induced a dramatic recovery of the enzymatic activity. This prompt and dynamic manipulation of DNAzyme activity further suggested a great controllability of this ligand-induced DNA structural regulation. Unlimited to the FC modification, treatment of CB[7] could also control the AD-modified DNAzyme (Supplementary Fig. 18), with relatively weaker efficiency on real-time regulation of DNAzyme activity (Supplementary Fig. 19), which was consistent with the slow kinetic binding/dissociation process on AD as observed above.

More than regulation of the functional DNA system in test tubes, we further wondered whether this ligand-controlled DNA hybridization could also occur inside living cells to manipulate biological processes. Previous investigations already suggested low cytotoxicity and excellent biocompatibility of CB[7] to ensure its wide applications in biological systems[50–53]. To this end, we selected the well-studied antisense oligonucleotide (ASO)[54] as the regulatory target, and checked how our ligand invasion could control the functionality of ASO on gene

expressions. The sequence of ASO was designed to target the mRNA of destabilized GFP (dsGFP), and phosphorothioate bonds were introduced in all the investigated ASOs to enhance their effects[55]. To directly manipulate its function, we placed three AD-modified nucleotides into this 18-nt oligonucleotide (see Supplementary Table 1 for the sequence information). Ideally, in the absence of invading ligand CB[7], the modified ASO would knockdown the expression of dsGFP through the formation of DNA/RNA hybrid; however, with the invasion of CB[7], the ASO would be unable to target the mRNA, resulting in the failure of dsGFP knockdown (Fig. 7a). The lentivirus infected HEK293T cells with stably expressed dsGFP were selected as the model cell line. Interestingly, with introduction of three AD-modified cytosine sites, the ASO still exhibited a strong repression on dsGFP as detected by flow cytometry, which was close to the unmodified ASO (Fig. 7b), demonstrating this guest integration presented limited impacts on DNA/RNA hybridization, and moreover, was likely tolerable to RNase H, given that the effect of ASO partially relies on the RNase H activity upon the formation of DNA/RNA hybrid[56]. To examine whether the invasion of CB[7] could block the functionality of the AD-modified ASO, we incubated the ASO with CB[7] first to ensure the complete formation of the host–guest complexation before transfection. Indeed, introduction of CB[7] totally abolished the gene knockdown activity of ASO (Fig. 7b), indicating that complexation of CB[7] and the AD-modified nucleoside could prevent the formation of nucleic acid hybridization in the cellular context. In addition to the pre-formed complex, we also transfect the cells with modified ASO first, followed by incubation with CB[7] to investigate the effect of ligand invasion during the cellular process. As described in Fig. 7c, when transfected cells were incubated with CB[7], the knockdown efficiency of modified ASO was decreased, implying the CB[7]-induced blockage of hybrid formation. Notably, real-time invasion of CB[7] inside living cells was relatively less efficient than pre-incubation, which might be attributed by the kinetically controlled invasion process of CB[7] with limited cellular uptake. Nevertheless, these cellular treatments suggested CB[7]-controlled nucleic acid hybridization could be utilized to manipulate ASO-mediated gene regulations in living cells.

Moreover, since complexation with CB[7] abolished the gene knockdown efficiency of the guest-modified ASO, competition with free guests would ideally recover its function as described in the reverse process of Fig. 7a. Herein, we prepared the blocked ASO with complexed CB[7] at the guest-modified nucleosides. After transfection, cells were treated with the free guest AD. Compared with the untreated cells, incubation with AD induced significant fluorescence knockdown, indicating the released ASO function (Fig. 7c). Although the effect of ASO could not be fully recovered possibly due to the slow kinetic dissociation behavior as observed above, this reversing activity still indicated the controllability of host–guest complexation on DNA/RNA hybridization. To exclude the possibilities that either treatment of CB[7] or the free AD molecule might influence the expression of dsGFP, we confirmed that these two ligands could hardly impact fluorescence signals of flow cytometry (Supplementary Fig. 20). Collectively, these results suggested that the ASO function could be controlled by CB[7]-mediated host–guest complexation in living cells.

## Discussion

In summary, this work developed a ligand-invasion pathway to regulate DNA hybridization based on host–guest interactions. We proposed a concept of recognition handle as the ligand binding site to disrupt Watson–Crick base pairs and induce the dissociation of DNA duplex structures. Taking CB[7] as the invading ligand and its guest molecules that were integrated into the $N^4$-position of cytosine or the $N^6$-position of adenine as recognition handles, we successfully achieved orthogonal and reversible manipulation of DNA duplex dissociation and its recovery. Moreover, we successfully applied this approach of ligand-controlled nucleic acid hybridization to achieve

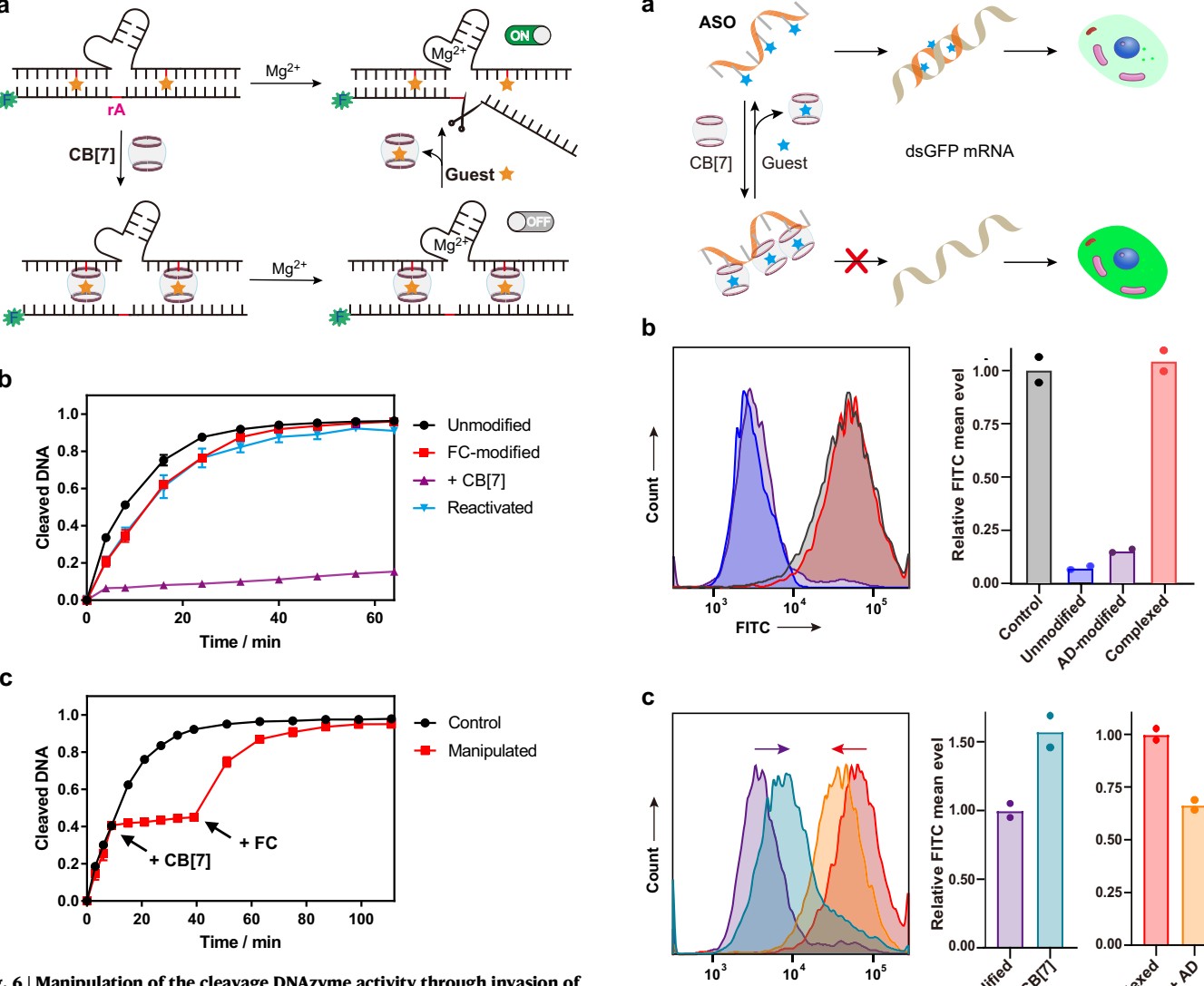

**Fig. 6 | Manipulation of the cleavage DNAzyme activity through invasion of CB[7]. a** Schematic figure for functional control of the magnesium-based DNAzyme system through the host–guest recognition. **b** Quantitative analysis of cleavage products by the FC-modified DNAzyme upon the treatment of CB[7] and the guest molecule FC. The unmodified DNAzyme system was utilized as control. The FC-modified DNAzyme presented a strong cleavage activity, whereas the treatment of CB[7] (300 μM) greatly inhibited its function. For the reactivated DNAzyme, the competing guest FC (500 μM) was incubated with the CB[7]-treated system to recover its cleavage activity. **c** Real-time deactivation and reactivation of the DNAzyme through the host–guest interaction. Quantitative analysis of cleavage products was measured at different time points. The FC-modified DNAzyme system without any treatment was utilized as control. The manipulated DNAzyme system was firstly treated with CB[7] (300 μM) at the 9-min point, and then mixed with the FC guest (500 μM) at the 39-min point. Data in (**b**) and (**c**) are presented as mean values with standard deviations derived from three independent replicates. Source data are provided as a Source Data file.

**Fig. 7 | Manipulation of the antisense oligonucleotide (ASO) to regulate gene expressions in living cells. a** Design of controllable functionality of ASO to regulate expressions of destabilized GFP in HEK293T cells. **b** Flow cytometry analysis of GFP expressions with treatment of different ASOs. The non-sense oligonucleotide was selected as control. The complexed ASO indicated the AD-modified oligonucleotide with CB[7] complexation. **c** Functional control of the AD-modified ASO by CB[7] and its reverse control of the complexed ASO by the AD guest. For the invasion of CB[7], the AD-modified ASO was selected as control; 200 μM CB[7] was added during cell culture to inhibit the function of the AD-modified ASO. For the competition with the free AD guest, the complexed ASO was selected as control; 100 μM AD was added during cell culture to release the antisense activity of ASO. In (**b**) and (**c**), the left panels showed the flow cytometry profiles, and the right panels showed the quantitative analysis of average fluorescence normalized to the control samples. Bar graphs showed the mean data derived from two independent replicates. Source data are provided as a Source Data file.

functional regulations of both the DNAzyme cleavage in test tubes and the antisense oligonucleotide in living cells. These results described a general strategy to manipulate DNA hybridization through ligand invasion and demonstrated its applicability in functional nucleic acid systems, which established a ligand-controlled approach to dynamically manipulate nucleic acid hybridization without any sequence constraints.

Our design of ligand invasion establishes a general pathway toward dynamic manipulation of nucleic acids through ligand invasion. Compared with the light irradiation, the ligand treatment exhibits

some superior advantages for specific scenarios, such as a deep reach into the inner part of a system (e.g. biological tissues) that light may not be accessible and a precisely dose-dependent control. One can envision that diversified nucleic acid nanostructures may be manipulated by chemical ligands for nucleic acid nanotechnologies, and kinds of functional nuclei acids may be ligand-controlled for cellular regulations on the basis of this concept, which would provide many possibilities for both nanomaterial developments and biological

regulations. Since nuclei acid hybridization typically relies on the formation of Watson–Crick base pairs, unlimited to DNA structures, it is also fully expectable that RNA duplexes or DNA/RNA hybrids would be manipulated in the same way through ligand invasion. More than the DNAzyme and ASO systems investigated in this work, other functional nucleic acids, such as ribozyme[57], siRNA[58] and guide RNA of CRISPR[59,60], may be manipulated similarly on the basis of this concept. Besides, given so many types of host–guest interactions beside the CB[7] system, one can also envision that kinds of supramolecular interactions may be employed to regulate nucleic acid structures through our design of recognition handles, which would pave the way for wide applications in a variety of nucleic acid-involving fields.

## Methods

### Materials

All solvents were commercially available and utilized without further purification. All unmodified DNA sequences used in this study were ordered from Sangon Biotech. The DNA phosphoramidites and CPG were purchased from DNA Chem. The HEK293T cell line was obtained from SUNNCELL (catalog number: SNL-015) and used without authentication testing. The compound masses were weighed on a microbalance with a resolution of 0.1 mg. TLC (Thin-layer chromatography) analysis was performed through pre-coated silica gel plates. Column chromatography carried out by silica gel (#100–200). $^1$H, $^{13}$C and $^{31}$P NMR spectra for compound characterizations (Supplementary Fig. 21–44) were recorded on Bruker AVANCE III 400 MHz spectrometer. The mass spectra (MS) were recorded on LCMS-2010A. UV absorbance was recorded by the Shimadzu 2600 UV-Vis spectrophotometer. Circular dichroism spectra were recorded by the Jasco J1700 CD Spectrometer. ITC experiments were carried on MicroCal iTC200 at 298 K. Gel shift was imaged by Gel Image System (Tanon 2500 R). The fluorescence was recorded by a SHIMADZU-RF-6000 fluorescence spectrophotometer. Flow cytometry was measured by the BD FACSCanto II flow cytometry instrument (BD Biosciences).

### Synthesis of modified ODNs

Guest-modified ODNs used in this study were prepared by the solid phase synthesis according to the conventional phosphoramidite protocol on a K&A H-8 DNA Synthesizer using the universal CPG. After the standard procedure of ODN synthesis, the products were cleaved from CPG, followed by treatment of aqueous ammonia (22%) at 55 °C for 12 h to remove the protecting groups. The crude ODNs were then purified by HPLC on the Agilent 1260 system using the PLRP-S column (250 mm × 4.6 mm, 100 Å, 5 μm) after ethanol precipitation. The aqueous solution of 0.1 M TEAA (triethylammonium acetate) with 50% CH$_3$CN was selected as buffer A, and the 0.1 M TEAA solution alone was selected as buffer B for the mobile phase. The ODNs were purified using a gradient of 5–60% buffer A over 50 min at a 1 mL/min flow rate. The purified products were collected, lyophilized, and then redissolved in water to remove salt by the Bio-Spin 6 column (Bio-Rad) before stored up at −20 °C for the following assays. ESI spectra of these modified ODNs were presented in Supplementary Fig. 45–53.

### Measurements of the duplex melting temperatures (T$_m$)

The FRET-based assays were conducted to determine T$_m$ values of duplex DNA. Briefly, 200 nM FAM-labelled ODN (see Supplementary Table 1) and 250 nM of the corresponding complementary oligonucleotides strands (cODN) labelled with BHQ-1 were annealed to form the duplex structure in the sodium cacodylate buffer (pH = 7.2, 20 mM sodium cacodylate, 100 mM NaCl and 100 μM EDTA) with different concentrations of CB[7]. Fluorescence changes were recorded by the Bio-Rad CFX96 Touch real-time fluorescence quantitative PCR (λex = 480 nm, λem = 520 nm) along with the increasing temperature from 4 °C → 95 °C at a rate of 0.5 °C/min. Data were analyzed by GraphPad

Prism 7.0. The T$_m$ value was determined by the temperature point of 50% conversion of the fluorescence intensity.

### Isothermal Titration Calorimetry (ITC) measurements

The association constants and thermodynamic parameters for the binding affinities of CB[7] with guest were determined by titration calorimetry (MicroCal iTC200) in the sodium phosphate buffer solutions (pH = 7.2, 20 mM sodium phosphate, 100 mM NaCl, 100 μM EDTA, containing 5% (v/v) DMSO for enhanced solubility of guest molecules). CB[7] (70–80 μM) was placed in the sample cell, to which BA or TB (0.8–1.25 mM) were added stepwise in a series of 25 or 30 injections, and the heat generated was recorded at 25 °C. For determination of the binding constants on AD and FC, the competitive method was utilized. Briefly, CB[7] (70–80 μM) was firstly mixed with BA (0.8 mM) in the sample cell, to which AD or FC (0.8–1.25 mM) were added stepwise in a series of 30 injections, and the heat generated was recorded at 25 °C. The data were analyzed and fitted by the Origin program (v7.0552).

Similarly, the binding affinities of CB[7] with guest-containing ODNs were also determined by ITC in the sodium phosphate buffer solutions (pH = 7.2, 20 mM sodium phosphate, 100 mM NaCl and 100 μM EDTA). Guest-containing ODNs (20–35 μM) was placed in the sample cell, to which CB[7] (500–600 μM) was added stepwise in a series of 32 injections, and the heat generated was recorded at 25 °C. The data were analyzed and fitted by the Origin program (v7.0552).

### UV-based kinetic analysis of CB[7] invasion

Kinetic measurements of the invasion of CB[7] into the duplex DNA were recorded by the Shimadzu 2600 UV-Vis spectrophotometer. 10 μM unmodified 15-nt ODN (control group) or the modified ODN with AD-integrated cytidine was annealed with 12 μM cODN in the sodium phosphate buffer (pH = 7.2, 20 mM sodium phosphate, 100 mM NaCl and 100 μM EDTA). The mixture was placed in a cuvette in the UV apparatus (Shimadzu 2600 UV-Vis spectrophotometer) for 20 min at 22 °C and then UV absorbance at 260 nm was recorded in a kinetic mode upon the addition of different concentrations of CB[7].

### Measurements of CD spectra for CB[7] treatment

Circular dichroism spectra for the invasion of CB[7] into the duplex DNA were recorded by the Jasco J1700 CD Spectrometer. Final concentrations of 10 μM unmodified/AD-modified 15mer and 19mer dsDNA was annealed with or without 500 μM CB[7] in the sodium phosphate buffer (pH = 7.2, 20 mM sodium phosphate, 100 mM NaCl and 100 μM EDTA). The mixtures were then placed in the CD cuvette for 20 min at 22 °C before spectral scans.

### Fluorescence measurements for the hairpin formation

To determine the conversion from duplex to the hairpin forms, 10 nM 19-nt AD-containing and 10 nM FC-containing ODNs were annealed with 20 nM Hp-1 (labelled by TMR and BHQ-2) and 20 nM Hp-2 (labelled by FAM and BHQ-1) in the presence of different concentrations CB[7] in the sodium phosphate buffer (pH = 7.2, 20 mM sodium phosphate, 100 mM NaCl and 100 μM EDTA). The fluorescence signals of FAM and TMR were scanned by a SHIMADZU-RF-6000 fluorescence spectrophotometer and collected with LabSolutions RF 1.11. (TMR: λex = 550 nm; FAM: λex = 480 nm).

### Fluorescence measurements for DNA duplex dissociation and recovery

The 15-nt modified ODN (20 nM, FAM-labelled strand) with either three AD-modified or three FC-modified cytidine sites was firstly annealed with the complementary strand (24 nM, BHQ1-labelled strand) in the sodium phosphate buffer (pH = 7.2, 20 mM sodium phosphate, 100 mM NaCl and 100 μM EDTA). After the formation of DNA duplex

structure, 5 μM CB[7] was added into the mixture at the room temperature (~22 °C). The sample was incubated at the room temperature for two hours before measurements of the FAM fluorescence. As comparison, 15-nt modified ODN without the complementary strand and the duplex DNA without treatment of CB[7] was processed through the same procedure and their fluorescence was also scanned as control.

For kinetically monitoring of DNA duplex dissociation and recovery, fluorescence of 15-nt modified ODN (50 nM, FAM-labelled strand) with either three AD-modified or three FC-modified cytidine sites was monitored in the sodium phosphate buffer (pH = 7.2, 20 mM sodium phosphate, 100 mM NaCl and 100 μM EDTA) at the room temperature (~22 °C). During the process of the kinetic monitoring, the complementary strand (62.5 nM, BHQ1-labelled strand), 10 μM CB[7] and 30 μM competing guest (AD or FC) were added into the mixture at the designated time points.

For cyclic treatments of CB[7] and FC, the 15-nt modified ODN (50 nM, FAM-labelled strand) with three FC-modified cytidine sites was firstly annealed with the complementary strand (62.5 nM, BHQ1-labelled strand) in the sodium phosphate buffer (pH = 7.2, 20 mM sodium phosphate, 100 mM NaCl and 100 μM EDTA). After the formation of DNA duplex structure, designated concentrations of CB[7] and FC were added into the mixture sequentially. After each addition, the mixture was incubated at the room temperature (~22 °C) for 60 min before the fluorescence measurement.

### Analysis of controllable 8-17-DNAzyme

To check whether the invasion of CB[7] could abolish the enzymatic activity of DNAzyme, the FC/AD-containing DNAzyme strand (0.5 μM) was incubated with the substrate strand (FAM-labelled with an rA site, 1 μM) in the reaction buffer (pH = 7.2, 20 mM Tris–HCl, 100 mM NaCl and 100 μM EDTA) in the presence or absence of 10 μM (for the AD-modified DNAzyme) or 300 μM (the FC-modified DNAzyme) CB[7] for 2 h. The cleavage was initiated upon the addition of 10 mM $Mg^{2+}$. Aliquots were taken at the designated time points and mixed with the quenching buffer containing 100 mM EDTA. The products were then analyzed by denaturing PAGE and visualized by Gel Image System (Tanon 2500 R). For the reactivated DNAzyme, 500 μM competing guest (FC/AD) was incubated with the CB[7]-treated system for two hrs (for FC-modified DNAzyme) or overnight (for the AD-modified DNAzyme) before addition of 10 mM $Mg^{2+}$. The unmodified DNAzyme system was also conducted as control.

For real-time manipulation of the DNAzyme system, the FC/AD-containing DNAzyme strand (0.5 μM) was incubated with the substrate strand (FAM-labelled with an rA site, 1 μM) in the reaction buffer (pH = 7.2, 20 mM Tris–HCl, 100 mM NaCl and 100 μM EDTA). The cleavage was initiated upon the addition of 10 mM $Mg^{2+}$. 10 μM CB[7] and 500 μM AD for the AD-modified DNAzyme, or 300 μM CB[7] and 500 μM FC for the FC-modified DNAzyme were then added into the reaction at the designate time points. Aliquots at different time points were taken during the process and mixed with the quenching buffer containing 100 mM EDTA. The products were then analyzed by denaturing PAGE and visualized by the Tanon 2500 R Gel Image System. Data were collected with Tanon MP (v1.0) and analyzed by Tanon Dots (v4.2). As comparison, the same DNAzyme system without treatment of CB[7] or FC/AD was also conducted the same procedure. These uncropped scans of gels were presented in Supplementary Fig. 54–57.

### Expressional regulation of cellular green fluorescent protein by antisense oligonucleotides

HEK293T-dsGFP cells were prepared according to a general procedure for lentiviral packaging. Briefly, the DNA sequence encoding destabilized GFP (dsGFP) was cloned into the pHR-SFFV (Addgene #79121) lentiviral vector to generate pHR-SFFV-dsGFP for dsGFP expression.

$1 \times 10^6$ HEK293T cells were seeded into six-well plates and co-transfected with 1 μg pMD2.G (Addgene #12259), 2 μg pCMV-dR8.2 (Addgene #84550) and 3 μg pHR-SFFV-dsGFP using Lipofectamine 3000. The transfection medium was replaced by a 2 mL fresh medium after a 6-hr incubation. 24-hr later, the culture medium was collected and treated with 1.6 μL polybrene (10 μg/μL) before infecting the freshly seeded HEK293T cells with a density of $2 \times 10^4$ cells in a six-well plate. The lentiviral infected cells were then collected and sorted by the flow cytometer (BeckmanmoFlo XDP) to obtain HEK293T-dsGFP cells.

One day prior to transfection, $4 \times 10^5$ HEK293T-dsGFP cells per well were seeded into a 6-well plate containing 0.5 ml complete medium (DMEM with 10%FBS). Cells were then transfected with ASOs (5′-GAGCTGCACGCTGCCGTC -3′, 1 μM) using a standard transfection protocol. For the complexed ASO, the AD-modified ASO was incubated with CB[7] (20 μM) to form host–guest complexation before transfection. Each ASO was tested in two separate wells as biological replication and processed independently. Briefly, each strand was incubated in 250 μL DMEM with 5 μL Lipofectamine 3000 for 15 min. The mixture was subsequently added dropwise into the cells. After a 4-hr incubation, the transfection medium was removed and changed to the fresh complete medium or the medium containing CB[7] (200 μM) or the AD guest (100 μM). Cells were then further cultured at 37 °C in 5% $CO_2$ for an additional 48 h.

Cultured cells were then harvested for determination of GFP expression by flow cytometry. In brief, each well was rinsed with 1× PBS and then dissociated with 0.25% trypsin-EDTA. Digestion was terminated by adding DMEM supplemented with 10% FBS. Cells were washed once with 1× PBS and then suspended in 1× PBS before filtering with a 35-μm cell strainer tube. A cell count was performed, and the sample was adjusted to 106 cells per mL. GFP levels of individual well were measured on the flow cytometer. Data collection was carried out using a BD FACSCanto II flow cytometry instrument (BD Biosciences). Data were collected with the BD FACSDiva software (v6.1.3) and analyzed by the FlowJo software (v7.2).

### Reporting summary

Further information on research design is available in the Nature Research Reporting Summary linked to this article.

## Data availability

The data that support the findings of this study are available within the article and Supplementary Information files, and are also available from the corresponding author upon request. Source data are provided with this paper.

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

## Acknowledgements

The authors acknowledge funding from National Natural Science Foundation of China (Nos. 21977122 and 22222706, L.X.) and the National Key R&D Program of China (2020YFA0211200, L.X.).

## Author contributions

L. Xu designed the project. L. Xiao and L.-L.W. carried out the experiments with assistance from C.-Q.W., H.L., Q.-L.Z. and Y.W., L. Xiao analyzed the data. L. Xu supervised this work and wrote the manuscript.

## Competing interests

The authors declare no competing interests.
