## [Peer Review File · Nature Communications]

Controllable DNA Hybridization by Host-Guest Complexation-Mediated Ligand InvasionREVIEWER COMMENTS

Reviewer #1 (Remarks to the Author):

Lin Xiao, et al here described an interesting approach to regulate nucleic acid hybridization. They introduced a mutual recognition site in nucleobase to mediate the competition between ligand invasion and base pairing, so that the duplex DNA could be dissociated or recovered by ligand control. Based on this concept, they selected the CB[7]-based host-guest system and placed different guests of CB[7] into the nucleobase as the binding site. They successfully achieved reversible and orthogonal manipulation of DNA hybridization by treatments of CB[7] and its guest molecules. Employing this design, they further demonstrated its applicability in functional control of the RNA-cleaving DNAzyme and the antisense oligonucleotide. Given the wide applications of dynamic nucleic acid technologies, this study will establish a general strategy for structural regulation of nucleic acids through ligand recognition. This represents a significant knowledge gap, given that the concepts presented in this study may also guide the design of more ligand-invading nucleic acid structures, and the relationship between nucleic acid structure and ligand design is poorly understood. Such knowledge is fundamental to the understanding of nucleic acid structure and ligand design. The objective of this project is to create an excellent approach to regulate nucleic acid hybridization by introducing a mutual recognition site in nucleobase to mediate the competition between ligand invasion and base pairing. Addressing this critical knowledge gap will promote the development of related research fields. The manuscript is well written, logically clear, has important scientific value, and should be of great interest to the readers. And the results are solid, well presented and the statistical analysis would help a lot for related readers. So, I highly recommend to publish it after some necessary revision.

1. Utilization of tens or hundreds of micromolar CB[7] to manipulate nucleic acid hybridization is totally fine outside the cellular environment. However, when applying in living cells, the toxicity of CB[7] needs to be considered. No toxicity tests or literature citations regarding the biocompatibility of CB[7] are included in the manuscript. Although the authors showed that 200 μM CB[7] seemed tolerable for HEK293T cells, would this dose of CB[7] be acceptable for cellular contexts in general?
2. ΔT_m values were listed in Supplementary Table 2 but not in Table 1 of the main text. These values should also be listed in Table 1 for a clear comparison.
3. The source or preparation of HEK293T-dsGFP cells should be provided.
4. There are many abbreviations in use. Please show the full names before use them.
5. There are some minor errors such as there are some grammar errors which should be revised carefully.
6. In the Supplementary Scheme 1, the chemical name "CEO(i-pr)2NPCl", in the figure 2e "(OCH₂CH₂CN)P(iPr₂N)₂," the abbreviation for the chemical group N,N-Diisopropyl should be the same. "(iPr)₂" or "iPr₂N".

Reviewer #2 (Remarks to the Author):

The manuscript by L. Xu and co-workers describes a novel approach towards supramolecular regulation of the DNA hybridization by complexation of CB7 with modified nucleobases. I find this method very interesting and relevant and also fitting to the readership of Nature Communications. The conclusions are well supported with the data. The description of the experimental procedures is detailed enough to be reproduced. However, I have several concerns regarding the positioning of the

results in comparison to the existing literature as well as presentation.

- 1) The authors should extend their literature search and consider other macrocyclic hosts, e.g. cyclodextrins, that also have been used for manipulation of DNA. One of the examples is the following: "Opening of DNA double helix at room temperature: Application of α -cyclodextrin self-aggregates" (<https://doi.org/10.1039/C0NR00184H>). The corresponding comparison with such systems + references should be added to the introduction.
- 2) Page 3, last paragraph: "...Although previous studies on CB[7] had introduced the host-guest interactions into nucleic acid molecules,²⁹⁻³³ these investigations did not involve direct regulation on the native structures of nucleic acids but only imposed external impacts to control their functionalities..." Unfortunately, I cannot completely agree with the authors. There are several examples of manipulation of nucleic acids structure by CB or CB-ligand complexes, although without direct complexation between CB and nucleic acids, e.g.: (a) chiral transition of DNA between B- and Z-forms (<https://doi.org/10.1002/adv.201800231>), (b) folding/unfolding of quadruplex DNA (<https://doi.org/10.1093/nar/gkx025>), (c) DNA condensation (for example, <https://doi.org/10.1038/srep04210>, <https://doi.org/10.1039/B704279E>). The authors should provide the honest comparison of their approach with the existing ones and also point out not only advantages but also drawbacks of the new approach they offer.
- 3) Page 5, first paragraph: the mentioned literature values for binding constants with CB7 were determined in water. At the same time, all measurements in this manuscript were performed in buffer solutions with high ionic strength and high concentration of sodium cations. For this reason, the comparison with the literature data is not relevant because sodium cations from the buffer bind to the carbonyl portals of CB7 and can significantly reduce the binding constants of the complexes with organic guests. Therefore, the binding constants of CB7 with the corresponding guests should be determined in the buffer.
- 4) In addition to the previous concern: the manuscript lacks quantitative data. In my opinion, it is necessary to determine the binding constant values with CB7 for all the modified oligonucleotides.
- 5) For the discussion of the orthogonal binding (page 8), the authors should provide clear ratios of oligos and CB7 for both binding equilibria.
- 6) Captions of figures and tables should contain information about the concentrations, it is not convenient to refer every time to the experimental section.
- 7) Minor point: the abbreviations Ba and Tb for 1,4-benzenedimethanamine and 4-tertbutylbenzylamine, respectively, can be easily confused with barium and terbium. I would recommend to choose different abbreviations.

Taking together, I think that the manuscript can be published after major revision.

Reviewer #3 (Remarks to the Author):

Xu et al. proposed reversible hybridization of DNA duplex by host-guest interaction. They modified adenine and cytosine with guest molecule (adamantane; Ad, ferrocene; Fc, and so on) that can form stable complex with host molecule (cucurbit[7]; CB[7]). First, they examined complexing behavior between the ODN involving guest molecule with CB[7]. Then, hybridization behaviors in the presence and absence of CB[7] were investigated, revealing efficient, clear-cut, and reversible regulation. Next, they applied this system to the reversible on-off regulation of DNAzyme reaction in test tube, which was successful. Finally, reversible gene-regulation was demonstrated in cell with Fc-modified ASO and CB[7], which was also successful, although switching was moderate probably due to the slow dissociation of CB[7].

Overall, this is a very nice study using guest-modified nucleobases and host for the reversible regulation of hybridization. All the results were convincing and necessary control experiments were adequately carried out. This is certainly another strategy of reversible hybridization. But compared this "recognition handle" concept with previous ones, it seems to me moderate advantages; 1) base-modification is necessary as photochromic nucleoside (Ref.20), 2) repetitive operation produces waste as toehold exchange (Ref.7), 3) reversible regulation natural DNA duplex by invasive photoswitchable ligand has been reported (Dohno et al., JACS, 2007, 129, 11898). I agree that this host-guest strategy is new, but compared with numerous hybridization control studies, I could not find

distinct novelty required for Nat. Commun.

1) Nucleic Acids Res., 2022, 50, 1241 (Host-guest chemistry for RNA hybridization)

This paper also uses the same Ad-CB[7] complexation to inhibit RNA hybridization. Here, Ad is attached at 2'-OH of ribose. The authors should cite this paper and describe the difference.

2) Cyclodextrin

beta-Cyclodextrin also forms strong complex with Ad. Why they did not examine Cyd, which seems suitable for in vivo application?

3) DNA/RNA hybridization

In addition to DNA duplex, the authors had better examine regulation of DNA/RNA hybridization, because modified ASO targets not DNA but RNA.

Point-By-Point Response

Reviewer #1 (Remarks to the Author):

“Lin Xiao, et al here described an interesting approach to regulate nucleic acid hybridization. They introduced a mutual recognition site in nucleobase to mediate the competition between ligand invasion and base pairing, so that the duplex DNA could be dissociated or recovered by ligand control. Based on this concept, they selected the CB[7]-based host-guest system and placed different guests of CB[7] into the nucleobase as the binding site. They successfully achieved reversible and orthogonal manipulation of DNA hybridization by treatments of CB[7] and its guest molecules. Employing this design, they further demonstrated its applicability in functional control of the RNA-cleaving DNAzyme and the antisense oligonucleotide. Given the wide applications of dynamic nucleic acid technologies, this study will establish a general strategy for structural regulation of nucleic acids through ligand recognition. This represents a significant knowledge gap, given that the concepts presented in this study may also guide the design of more ligand-invading nucleic acid structures, and the relationship between nucleic acid structure and ligand design is poorly understood. Such knowledge is fundamental to the understanding of nucleic acid structure and ligand design. The objective of this project is to create an excellent approach to regulate nucleic acid hybridization by introducing a mutual recognition site in nucleobase to mediate the competition between ligand invasion and base pairing. Addressing this critical knowledge gap will promote the development of related research fields. The manuscript is well written, logically clear, has important scientific value, and should be of great interest to the readers. And the results are solid, well presented and the statistical analysis would help a lot for related readers. So, I highly recommend to publish it after some necessary revision.”

Response: We highly appreciate the reviewer's positive comments.

“1. Utilization of tens or hundreds of micromolar CB[7] to manipulate nucleic acid hybridization is totally fine outside the cellular environment. However, when applying in living

cells, the toxicity of CB[7] needs to be considered. No toxicity tests or literature citations regarding the biocompatibility of CB[7] are included in the manuscript. Although the authors showed that 200 μ M CB[7] seemed tolerable for HEK293T cells, would this dose of CB[7] be acceptable for cellular contexts in general? ”

Response: We highly appreciate the reviewer’s point regarding the toxicity of CB[7]. Several cellular studies have shown that CB[7] exhibits very low cytotoxicity at sub-mM concentrations or even up to 1 mM concentrations (*PLoS One*, 2010, 5, e10514; *Org Biomol Chem*, 2010, 8, 2037). For instance, CB[7] has shown high cell tolerance at concentrations of up to 1 mM in cell lines originating from the kidney (HEK293), liver (HepG2) or blood (RAW 264.7) tissue (*PLoS One*, 2010, 5, e10514); when incubated with the CHO-K1 cell line for 2 days, the IC₅₀ value of CB[7] was found to be 0.53 mM (*Org Biomol Chem*, 2010, 8, 2037). Moreover, the biocompatibility of CB[7] was also previously reported through *in vivo* studies on mice, where intravenous administration of CB[7] showed a maximum tolerated dosage of 250 mg kg⁻¹ (*Org Biomol Chem*, 2010, 8, 2037). Another study using mice as subjects also suggested no significant difference for the body weight change among the groups of mice after administration of a single dose of 5 g/kg orally, 500 mg/kg peritoneally and 150 mg/kg intravenously (*Sci Rep*, 2018, 8, 8819). Collectively, these previous investigations explicitly suggest low cytotoxicity and excellent biocompatibility of CB[7] to ensure wide applications of CB[7] in biological systems. We have included the description regarding the toxicity of CB[7] and related references (Ref 50-53) into the revised manuscript.

“2. ΔT_m values were listed in Supplementary Table 2 but not in Table 1 of the main text. These values should also be listed in Table 1 for a clear comparison.”

Response: We appreciate the reviewer’s suggestion. In the revised manuscript, ΔT_m values have been included in Table 1.

“3. The source or preparation of HEK293T-dsGFP cells should be provided.”

Response: HEK293T-dsGFP cells were prepared according to a general procedure for lentiviral packaging as described below.

“HEK293T-dsGFP cells were prepared according to a general procedure for lentiviral packaging. Briefly, DNA sequence encoding destabilized GFP(dsGFP) was cloned into the pHR-SFFV (Addgene #79121) lentiviral vector to generate pHR-SFFV-dsGFP for constitutive expression of dsGFP. To obtain viral production cells, 1, 000, 000 cells were seeded into six-well plates before co-transfection of 1 µg pMD2.G (Addgene #12259), 2 µg pCMV-dR8.2 (Addgene #84550) and 3 µg pHR-SFFV-dCas9-VPR-EGFP using 16 µL P3000 and 7 µL Lipofectamine 3000. The transfection medium was then changed to 2 mL fresh medium after 6-hr incubation. Twenty-four hours later, the culture medium was collected and filtered to remove cells, and treated with 1.6 µL polybrene (10 µg/µL) before infecting the freshly seeded 293T cells with a density of 20, 000 cells in a six-well plate. The infected cells were then collected and sorted by flow cytometer (BeckmanmoFlo XDP) to obtain HEK293T-dsGFP cells.”

These experimental details have been included in *Methods* in the revised manuscript.

“0. *There are many abbreviations in use. Please show the full names before use them.*”

Response: We have checked all the abbreviations and showed the full names when they are used for the first time.

“1. *There are some minor errors such as there are some grammar errors which should be revised carefully.*”

Response: We have carefully revised and polished the manuscript thoroughly.

“2. *In the Supplementary Scheme 1, the chemical name “CEO(i-pr)2NPCI”, in the figure 2e “(OCH2CH2CN)P(iPr2N)2,” the abbreviation for the chemical group N,N-Diisopropyl should be the same. “(iPr)2” or “iPr2N”.*”

Response: This inconsistency on abbreviation has been revised.

Reviewer #2 (Remarks to the Author):

“The manuscript by L. Xu and co-workers describes a novel approach towards supramolecular regulation of the DNA hybridization by complexation of CB7 with modified nucleobases. I find this method very interesting and relevant and also fitting to the readership of Nature Communications. The conclusions are well supported with the data. The description of the experimental procedures is detailed enough to be reproduced. However, I have several concerns regarding the positioning of the results in comparison to the existing literature as well as presentation.”

Response: We highly appreciate the reviewer’s positive comments, and we have also clearly responded to the concerns as described below.

- 1) *The authors should extend their literature search and consider other macrocyclic hosts, e.g. cyclodextrins, that also have been used for manipulation of DNA. One of the examples is the following:” Opening of DNA double helix at room temperature: Application of α -cyclodextrin self-aggregates” (<https://doi.org/10.1039/C0NR00184H>). The corresponding comparison with such systems + references should be added to the introduction.”*

Response: We thank the reviewer for the valuable comments. The reviewer mentioned an interesting study on manipulation of duplex DNA by cyclodextrins, in which the nano-assemblies of cyclodextrins with hydroxyl-rich surfaces were capable of perturbing hydrogen bonds between base pairs. In fact, these effects of cyclodextrins may be imposed into any strands of DNA double helices without any sequence specificity, which are more like the performance of denaturing reagents rather than the strand-specific manipulation of desired DNA hybridization in our work. Besides, the high concentrations of cyclodextrins also make the reversible regulation of DNA duplex highly difficult. We have cited this reference (Ref 12) and another similar report (Ref 13), and discussed their disadvantages on controllable DNA hybridization in the introduction part.

- 2) *Page 3, last paragraph: “...Although previous studies on CB[7] had introduced the host-4*

guest interactions into nucleic acid molecules,29-33 these investigations did not involve direct regulation on the native structures of nucleic acids but only imposed external impacts to control their functionalities...” Unfortunately, I cannot completely agree with the authors. There are several examples of manipulation of nucleic acids structure by CB or CB-ligand complexes, although without direct complexation between CB and nucleic acids, e.g.: (a) chiral transition of DNA between B- and Z-forms (<https://doi.org/10.1002/adv.201800231>), (b) folding/unfolding of quadruplex DNA (<https://doi.org/10.1093/nar/gkx025>), (c) DNA condensation (for example, <https://doi.org/10.1038/srep04210>, <https://doi.org/10.1039/B704279E>). The authors should provide the honest comparison of their approach with the existing ones and also point out not only advantages but also drawbacks of the new approach they offer.”

Response: We highly appreciate the reviewer’s comments regarding manipulation of nucleic acids structure by cucurbiturils. The examples mentioned by the reviewer actually represent the indirect influence of cucurbiturils on nucleic acid structures. These approaches may be effective to alter nucleic acid structures, but their impacts are originated from specific reagents or assemblies, but not directly from cucurbiturils. In our design, we aim to manipulate nucleic acid hybridization directly by CB[7], which would make structural and functional regulation of nuclei acids more controllable. Inevitably, efforts on synthesis of chemically modified nucleic acids are needed to introduce the “recognition handle”, which may not be easy for sample preparation. In the revised manuscript, we have included these references (Ref 31, 32, 36 and 37) and more sentences to introduce the indirect influence of CB[7] on nucleic acid structures as followings.

“Although previous studies on cucurbiturils had introduced the host-guest interactions into nucleic acid manipulation, the role of cucurbiturils did not involve direct regulation on the native structures of nucleic acids but only imposed indirect or external impacts to alter their structural conversions or functional performance. For instance, some ligands that are capable of interacting with nucleic acids can be designed to be complexed with cucurbiturils to control their effects on nucleic acid structures, such as the B-Z conversion of duplex DNA, formation

of G-quadruplex and induction of DNA condensation, but these impacts are originated from specific reagents or assemblies, whereas not directly from cucurbiturils.”

- 3) *Page 5, first paragraph: the mentioned literature values for binding constants with CB7 were determined in water. At the same time, all measurements in this manuscript were performed in buffer solutions with high ionic strength and high concentration of sodium cations. For this reason, the comparison with the literature data is not relevant because sodium cations from the buffer bind to the carbonyl portals of CB7 and can significantly reduce the binding constants of the complexes with organic guests. Therefore, the binding constants of CB7 with the corresponding guests should be determined in the buffer.”*

Response: We highly appreciate the reviewer’s valuable suggestion. In our revised manuscript, binding constants of CB[7] with all the four guest molecules have been determined by ITC in the phosphate buffer (20 mM, pH = 7.2) with 100 mM NaCl. As shown in the new **Supplementary Fig. 1**, the binding constants are $1.64 \times 10^9 \text{ M}^{-1}$ for AD, $4.70 \times 10^8 \text{ M}^{-1}$ for FC, $8.88 \times 10^5 \text{ M}^{-1}$ for BA and $3.98 \times 10^4 \text{ M}^{-1}$ for TB. Indeed, the binding affinities between CB[7] and these guest molecules are greatly weakened compared with the previously reported results in water with acidic pH values. We have included these results in the revised manuscript and made the explanation in the main text.

- 4) *In addition to the previous concern: the manuscript lacks quantitative data. In my opinion, it is necessary to determine the binding constant values with CB7 for all the modified oligonucleotides.”*

Response: We appreciate the reviewer’s suggestion. In the previous version of the manuscript, we determined the binding constants between CB[7] and the cytosine-modified oligonucleotides (**Supplementary Fig. 2**). In this revised version, we have further included the binding constants between CB[7] and the adenine-modified oligonucleotides as shown in **Supplementary Fig. 3**. Consistently, CB[7] can specifically bind with the AD ($K_a = 1.34 \times 10^6 \text{ M}^{-1}$) and FC ($K_a = 7.44 \times 10^5 \text{ M}^{-1}$) groups of the modified adenine in the context of DNA strand but exhibited feeble affinities on the BA and TB groups. Notably, the binding affinities of CB[7]

on the modified adenine are relatively weaker than the modified cytosine, which may be attributed to different steric hindrance by the nucleobase. These quantitative data have been included into the revised manuscript.

5) *For the discussion of the orthogonal binding (page 8), the authors should provide clear ratios of oligos and CB7 for both binding equilibria.*

Response: We appreciate the reviewer's point. In **Figure 4**, the concentration of duplex DNA (10 nM) was far more less than CB[7] (50 nM ~ 100 μ M). Given the concentration-dependent binding behaviors between CB[7] and the guest-modified DNA, a low concentration of CB[7] (< 200 nM) could only dissociate the duplex DNA integrated with the AD guest, whereas a high concentration of CB[7] (> 200 nM) could dissociate both duplex structures. Therefore, the ratios between CB[7] and oligos are not essential in our conditions. For a clear understanding, we have included the information on the concentrations of DNA and CB[7] in the revised figure captions.

6) *Captions of figures and tables should contain information about the concentrations, it is not convenient to refer every time to the experimental section.*

Response: We highly appreciate the reviewer's important suggestions. In the revised manuscript, we have included the concentration information in the captions of figures and tables.

7) *Minor point: the abbreviations Ba and Tb for 1,4-benzenedimethanamine and 4-tertbutylbenzylamine, respectively, can be easily confused with barium and terbium. I would recommend to choose different abbreviations.*

Taking together, I think that the manuscript can be published after major revision.

Response: We have changed the abbreviations for these four guests into capital letters to avoid any confusion.

Reviewer #3 (Remarks to the Author):

“Xu et al. proposed reversible hybridization of DNA duplex by host-guest interaction. They modified adenine and cytosine with guest molecule (adamantane; Ad, ferrocene; Fc, and so on) that can form stable complex with host molecule (cucurbit[7]; CB[7]). First, they examined complexing behavior between the ODN involving guest molecule with CB[7]. Then, hybridization behaviors in the presence and absence of CB[7] were investigated, revealing efficient, clear-cut, and reversible regulation. Next, they applied this system to the reversible on-off regulation of DNzyme reaction in test tube, which was successful. Finally, reversible gene-regulation was demonstrated in cell with Fc-modified ASO and CB[7], which was also successful, although switching was moderate probably due to the slow dissociation of CB[7]. Overall, this is a very nice study using guest-modified nucleobases and host for the reversible regulation of hybridization. All the results were convincing and necessary control experiments were adequately carried out. This is certainly another strategy of reversible hybridization. But compared this “recognition handle” concept with previous ones, it seems to me moderate advantages; 1) base-modification is necessary as photochromic nucleoside (Ref.20), 2) repetitive operation produces waste as toehold exchange (Ref.7), 3) reversible regulation natural DNA duplex by invasive photoswitchable ligand has been reported (Dohno et al., JACS, 2007, 129, 11898). I agree that this host-guest strategy is new, but compared with numerous hybridization control studies, I could not find distinct novelty required for Nat. Commun.”

Response: We appreciate the reviewer’s positive comments as well as constructive criticisms. Herein, we would like to make some explanations regarding the reviewer’s concern. In facts, other than the widely developed light control, it still lacks a general pathway to reversibly disrupt a random nucleic acid duplex structure and regulate its hybridization by chemical ligands. Compared with the light irradiation, the ligand treatment exhibits some superior advantages for specific scenarios, such as a deep reach into the inner part of a system (e.g. biological tissues) that light may not be accessible and a precisely dose-dependent control. Besides, given that the recognition handle can be placed into either adenine or cytosine, any sequence of nucleic acid hybridization can be theoretically manipulated by ligand treatments

based on our design. The reviewer mentioned a previous work to regulate DNA duplex by a photo-switchable ligand (*JACS*, 2007, *129*, 11898.), but this approach can only work in a mismatched duplex with a constrained sequence, which is unable to be generalized for a random sequence of nucleic acid. Our design actually establishes a universal pathway to control nucleic acid hybridizations by chemical ligands without any sequence constraints. Moreover, the concept we proposed (“recognition handle”) is not limited to the host-guest system, but may be employed for molecular recognition in general and guide the design of more controllable nucleic acid structures by diversified ligand invasions.

“1) *Nucleic Acids Res.*, 2022, *50*, 1241 (*Host-guest chemistry for RNA hybridization*)

This paper also uses the same Ad-CB[7] complexation to inhibit RNA hybridization. Here, Ad is attached at 2'-OH of ribose. The authors should cite this paper and describe the difference.”

Response: This paper had already been cited in the previous version of this manuscript (ref 32 in the previous version or ref 39 in the revised version). We would like to point out that this previous study introduced the adamantane guest into the 2'-OH of ribose, which did not significantly influence the nucleic acid structures by CB[7] complexation but imposed steric impacts on molecular interactions between RNA and its binding protein, so that its functionality could be modulated. This approach is distinctly different from our design on direct regulation of nucleic acid hybridization. In consideration of this work and other relevant reports as mentioned by Reviewer 2, we have included the following sentence in the revised manuscript: “Although previous studies on cucurbiturils had introduced the host-guest interactions into nucleic acid manipulation, the role of cucurbiturils did not involve direct regulation on the native structures of nucleic acids but only imposed indirect or external impacts to alter their structural conversions or functional performance.”

“2) *Cyclodextrin*

beta-Cyclodextrin also forms strong complex with Ad. Why they did not examine Cyd, which seems suitable for in vivo application?”

Response: We appreciate the reviewer for mentioning another possible host molecule for

structural manipulation of nucleic acids based on our design. It would be interesting to confirm whether β -cyclodextrin could perform similarly as CB[7] on regulation of nucleic acid hybridization, and then check its potential applications on modulation of functional nucleic acids. This host ligand, although out of the current scope of this work, definitely deserves further investigations in the future studies.

“3) DNA/RNA hybridization

In addition to DNA duplex, the authors had better examine regulation of DNA/RNA hybridization, because modified ASO targets not DNA but RNA.”

Response: We appreciate the reviewer’s point. Even though we mainly focus on regulation of DNA duplex structures, the concept proposed in this work actually aims at regulation of Watson-Crick base pairs. Since nucleic acid hybridization typically relies on the formation of Watson-Crick base pairs, it is fully foreseeable that either DNA duplexes, RNA duplexes or DNA/RNA hybrids would be manipulated similarly as we have investigated in this work. We have included this explanation into the conclusion part of the revised manuscript.

REVIEWERS' COMMENTS

Reviewer #1 (Remarks to the Author):

The authors here described an interesting approach to regulate nucleic acid hybridization. The authors have carefully revised their manuscripts and answer the reviewer's questions. The manuscript is well written, has important scientific value, and should be of great interest to the readers. And the results are well presented and the statistical analysis would help a lot for related readers. Overall, it is an important study, and highly recommend to be published.

Reviewer #2 (Remarks to the Author):

After revision, the manuscript was significantly improved. I thank the authors for addressing all my concerns and for doing additional experiments and explanations. In my opinion, the manuscript can now be accepted for publication.